# Transition in social risk factors and adolescent motherhood in low- income and middle-income countries: Evidence from Demographic and Health Survey data, 1996–2018

M. Mamun Huda [1,2] *, Jocelyn E. Finlay [3], Martin O'Flaherty [1,2], Abdullah Al Mamun [1,2]

**1** Institute for Social Science Research, The University of Queensland, Brisbane, Queensland, Australia, **2** ARC Centre of Excellence for Children and Families over the Life Course, The University of Queensland, Brisbane, Queensland, Australia, **3** Department of Global Health and Population, Harvard TH Chan School of Public Health, Boston, Massachusetts, United States of America

* m.huda@uqconnect.edu.au

**Data Availability Statement:** The data underlying the results presented in the study are available from The DHS Program at https://dhsprogram.

## Abstract

Understanding the dynamics of social risk factors in the occurrence of adolescent motherhood is vital in designing more appropriate prevention initiatives in low-income and middle-income countries (LMICs). We aimed this study to examine the transition of social risk factors and their association with adolescent motherhood in LMICs since the initiation of the MDGs. We analysed 119967 adolescent girls (15–19 years) from 40-nationally representative Demographic Health Surveys in 20 LMICs that had at least two surveys: a survey in 1996-2003(baseline, near MDGs started) and another in 2014-2018(endline). Adolescent motherhood (having a live birth or being pregnant before age 20) was the outcome of interest, whereas social risk factors including household wealth, girls' level of education, and area of residence were the exposures. The association between adolescent motherhood and the social risk factors, as well as changes in the strength of the association over time were observed using multilevel logistic regression analysis. On an average, the proportion of adolescent mothers without education decreased by -15·61% (95% CI: -16·84, -14·38), whereas the poorest adolescent mother increased by 5·87% (95% CI: 4·74, 7·00). The national prevalence of adolescent motherhood remained unchanged or increased in 55·00% (11/20) of the studied countries. Comparing baseline to endline, the overall adjusted odds ratio (AOR) of adolescent motherhood increased for both poorest (AOR = 1·42, 95% CI: 1·28, 1·59) and rural residences (AOR = 1·09, 95% CI: 1·01, 1·17), and decreased, but not statistically significant for the low level of education (AOR = 0·92, 95% CI: 0·84, 1·01 for no education). Our study concludes that social risk factors of the adolescent mother had shifted in different directions during MDGs and SDGs eras, and adolescent mothers remained more disadvantaged than non-mothers in LMICs. Efforts need to be enhanced to improve adolescent girls' education. Intervention should be prioritised in disadvantaged communities to delay adolescent first birth and prevent adolescent motherhood in LMICs.

com/data/new-user-registration.cfm. Data are accessible free of charge upon a registration with the Demographic and Health Survey program (The DHS Program).

**Funding:** This research did not receive any specific grant. Authors were individually supported by the several grants; however, these sponsors were not involved to any of the study activities. MMH was supported by the RTP scholarship for PhD study funded by the Commonwealth Government of Australia and The University of Queensland. MO was supported by the ARC Centre of Excellence for Children and Families over the Life Course (Grant no: CE140100027), JEF's contribution was supported by the Dutch National Science Foundation, NWO-WOTRO (Grant no. W 08.560.002), and AM was supported by the National Health and Medical Research Council (Grant no: APP1083456). The funders had no role in study design, data collection and analysis, decision to publish, or preparation of the manuscript.

**Competing interests:** We declare no competing interests

## Introduction

Adolescent motherhood is a global concern, and the prevalence of adolescent motherhood is exceptionally high in low-income and middle-income countries (LMICs) [1, 2]. According to World Health Organization (WHO), an estimated 21 million girls aged 15–19 years in LMICs become pregnant, and about 12 million give birth every year [1]. In some LMICs, the overall prevalence of adolescent motherhood has declined; however, the rate of decline is slow, and the prevalence remains high in many countries, particularly in sub-Saharan Africa [2].

Adolescent pregnancy and childbirth have adverse health and wellbeing consequences for mothers and their children. Adolescent girls have a high risk of maternal mortality and morbidity due to pregnancy complications and unsafe abortion [3, 4]. Children born to adolescent mothers have an increased risk of premature birth, death, malnutrition, and low physical and mental development [5, 6]. Moreover, social consequences of adolescent motherhood include school dropout and failure to enter decent works (due to their double burden of household maintenance and child-rearing), and in some settings, violence, including suicide and homicide [7]. Considering adolescent motherhood's health and social consequences, prevention of adolescent pregnancy is a long-term global target, particularly since the Declaration of the Millennium Development Goals (MDGs) and now in the Sustainable Development Goals (SDGs) era. Prevention of adolescent pregnancy through improving access to sexual and health care services is an important issue within the 2030 UN SDGs and the UN Global Strategy for Women's, Children's, and Adolescent's Health [8, 9]. However, many adolescent health needs, including sexual and reproductive health, continue to be overlooked [10, 11]. Governments in many LMICs remain reluctant to move beyond the sexual and reproduction health right program of abstinence messaging for unmarried adolescents, and child marriage remains acceptable (or even encouraged) in many countries. Legal restrictions on emergency contraception and abortion in many LMICs are barriers to avoiding unwanted pregnancies [12]. Furthermore, adolescent friendly health systems for sexual health, family planning, and maternal health are lacking in many LMICs [13]. As such, the prevalence of adolescent motherhood in LMICs remains high [2].

Understanding factors associated with the prevalence of adolescent motherhood in LMICs enables the design of more targeted prevention strategies. Factors related to the adolescent pregnancy and motherhood are well studied in many LMICs [14–16]. Those studies reported various types of risk and protective factors, including social (i.e., sociocultural, environmental, and economic factors), behavioural (sexual risk behaviour, excessive use of alcohol, substance abuse), health service-related (i.e., inadequate, and unskilled health workers, long waiting time and lack of privacy at clinics, cost and misconceptions about contraceptives, and non-friendly adolescent reproductive services) factors, etc. [17, 18]. The social factors of adolescent pregnancy and motherhood may also operate at different levels such as individual, family or school, community, and national level [19, 20]. Factors that examined adolescent pregnancy and motherhood were mostly focused on the individual level, limited focused on the family and community-level factors, and few focused on national or country contexts in the literature [17, 21].

Individual level social risk factors such as poverty, lower educational level, and rural area of residence are some of the commonly reported risk factors of adolescent pregnancy in LMICs [22–25]. Pradhan R *et al.* conducted a systematic review study to understand factors associated with adolescent pregnancy in LMICs [24]. The review reported that limited education and low socioeconomic status were generally a risk for pregnancy among adolescent girls despite variations in study methodology. Living in a rural area is also reported as a risk factor for early pregnancy in some studies [24]. An empirical study conducted in the sub-Saharan African region

reported that girls' education and household wealth status were consistently associated with early motherhood in all five studied countries. The study estimated that adolescent having secondary or above education and belonging to the richest household wealth quintile reduced the odds of adolescent pregnancy by 63–67% and 53–68%, compared to their counterpart [25]. Studies in the Latin American region reported that adolescent first births continue to be more common among the poorest and rural residence [14, 23]. Those factors, of lower education and household wealth and rural residency, were also reported as primary risk factors in the south Asian region [22].

Although many country-specific and several review studies have been done, there is a lack of multicounty studies that would enable clear comparisons of those social risk factors associated with the occurrence of adolescent motherhood across countries in different regions. Furthermore, the transition of social risk factors of adolescent motherhood has not been focused enough in LMICs. Significant changes in the socioeconomic and demographic status have occurred in LMICs over the past twenty years, which may result in changes in the association between those social risk factors and the occurrence of adolescent motherhood in LMICs. For example, the primary school net enrolment rate in the developing regions reached 91% in 2015, up from 83% in 2000 [26]. Globally, the number of people living in extreme poverty has declined by more than half, falling from 1.9 billion in 1990 to 836 million in 2015. Most progress has occurred since 2000 [26]. Thus, assessing the transition of social risk factors of adolescent girls and occurrence of adolescent motherhood could help to understand vulnerable group for prioritising interventions to prevent adolescent motherhood in LMICs.

In this study, we assessed change in three social risk factors that are commonly linked with adolescent motherhood: household socioeconomic status, girls' level of education, and place of residence. We also examined change in the strength of the association between earliest (around the commencement of the MDGs) to recently (2014–2018). We employed a consistent methodology across 40 different surveys in 20 LMICs, allowing for a rigorous and transparent comparison of those social risk factors associated with adolescent motherhood across LMICs.

## Methods

### Ethics statement

The Demographic and Health Survey (DHS) data collection procedures were reviewed and approved by the ICF Institutional Review Board (IRB) [27]. Each survey is approved by the relevant country-specific ethical review board that oversees research studies on human subjects in each of the participated countries [27]. This study was also approved by the Human Ethics Research Office, The University of Queensland, Australia, Approval number: 2019001820.

### Data source, study design and participants

We used data from the nationally representative DHS in in this study. The DHS is a cross-sectional survey based on a multi-stage stratified sampling design. In DHS, usually, countries were divided into sub-national regions mostly based on local administrative boundaries. In each region, the population were further stratified according to urban and rural area residence (known as strata). Within these strata, enumeration areas mostly known as clusters were identified based on the most recent population census. At the first stage, these primary sampling units were selected based on probability proportional to the population size from each stratum. Complete household listing was made for each of the selected clusters. At the second stage, approximately 30–40 and 20–25 households were selected by equal probability systematic sampling in the selected clusters from rural and urban areas respectively [28]. The survey collects data on sociodemographic indicators and population health, including women's reproductive

and child health [29]. ICF, a global consulting and technology services company administers the DHS and provides identical core questionnaires and intensive training to each participating country to ensure quality, standardisation and comparability of the survey data across countries [30]. DHS cover six different geographic regions; sub-Saharan Africa, North Africa, west Asia or Europe, Central Asia, South and southeast Asia, Oceania, and Latin America and the Caribbean. The list of countries, regions, years and sample sizes for the interviewed women in this study is provided in S1 Table.

The target population in this study was adolescent girls aged 15–19 years old. Following the sample selection criterion of the study, a total of 119,967 adolescent girls (15–19 years) from 20 countries were selected, where 41570 were from baseline and 78397 were from endline survey. Among them, 21·01%(n = 8766) and 19·70% (n = 15410) were adolescent mothers in baseline and endline, respectively (S1 Fig). The sample selection criterion was, country with at least two all-women surveys: a survey in 1996–2003, near to MDG started (as baseline) and another survey in 2014–2018, recent time (as endline) so that the duration for assessing the transition of social factors can be nearly 15 years for all the studied countries. For the 20 selected countries, the number of surveys highly varied between baseline and endline surveys. Thus, for the consistency in the number of surveys, only two surveys (baseline and endline) were included in the study.

## Outcome measurement

The primary outcome of interest in this study was adolescent motherhood. We defined adolescent motherhood for women aged 15–19 as having either given birth or being currently pregnant at the time of the interview [29] (S2 Table).

## Measurement of social risk factors

The socioeconomic status of the adolescent girls was classified into five different categories: poorest, poor, middle, richer and richest based on a proxy measure of the wealth quintile of the girl's household. All interviewed households were ranked into five wealth quintiles based on the estimated wealth index within each survey. The wealth index is calculated using principal component analysis on a household's ownership of selected assets data, such as televisions and bicycles; materials used for housing construction; and types of water access and sanitation facilities [31]. Level of education in the following categories: no education, primary, and secondary or higher. The area of residence where the respondent was living at the time of the interview were categorised as either urban or rural (S2 Table). All three factors were consistently measured across the country and surveys in DHS [29].

## Statistical analysis

At first, we calculated the proportion of adolescent motherhood and the proportion of social risk factors among adolescent mothers and adolescent girls who are not mother across two time periods: baseline and endline. For notation, we will call non-mothers for the group of adolescents who are not mothers. Then, we estimated the country-specific population-weighted prevalence of adolescent motherhood at baseline and endline survey to see changes in the national level prevalence of adolescent motherhood over the study period. The DHS sampling frame is designed such that with weighting, prevalence can be nationally (and by urban/rural strata and region) representative [29]. The weights are the inverse of the probability of selection and the response rates (individual/households) for women in the stratum. Secondly, we explored changes in the proportion of social risk factors between baseline and endline for the group of adolescent mothers and non-mothers. Finally, we performed a

regression analysis to identify the adjusted association between the social risk factors and the prevalence of adolescent motherhood and see how the association changed over time in LIMCs.

In the DHS survey design (multistage cluster sampling), interview participants are usually nested within each country's survey cluster (at a higher level). When we pooled the study countries for our pooled analysis, clusters were further nested within countries. Finally, in the pooled analysis, study participants were nested at three hierarchical levels: country, cluster, and individual. Due to this hierarchical structure of the data, study participants who nested at different levels correlate with one another. Failing to address the correlation may underestimates the standard error, leading to an overstatement of statistical significance. The multilevel regression model can handle this type of correlated data and produce unbiased estimates. Thus, following the survey design and type of outcome measures, the multilevel logistic regression model was used to identify the independent factors associated with adolescent motherhood by adjusting the correlation in data due to grouping at cluster level within the country and country level in pooled data [32, 33].

In the multilevel regression analysis, we developed three different regression models. Model 1 examines the relationship between the outcome and a single variable without considering adjustment for other variables. Model 2 adjusts for all three social risk factors. Model 3 extended Model 2 by including two–way interactions between the social risk factors and survey time (baseline/endline) and the other two social risk factors. A 3-level (at country, cluster with the country, and individual) logistic regression model was developed in the pooled analysis. In contrast, it was a 2-level (at cluster within the country, and individual) logistic regression model for country-specific analysis. All data were analysed in Stata/MP 16.1 (StataCorp, College Station, Texas) [34].

The odds ratio (OR) and adjusted odds ratio (AOR) generated from the exponent of β-coefficient of the unadjusted and adjusted model, respectively, and corresponding 95% CI were reported. AOR attributed from the interaction term in model 3 represents changes in the effect of the social risk factors on adolescent motherhood (from baseline to endline). AOR>1 (<1) for the interactions terms in Model 3 represents increased (decreased) in the association between adolescent mother and social risk factors at endline compared to baseline.

## Results

Concerning our studied social risk factors, we found that social risk factors of the adolescent mother had shifted in different directions over time. Overall, about 50% of the mothers were either poorest or poorer, 31% were without educations, or 72% lived in rural areas, whereas it was about 36%, 14% and 59% respectively for non-mother (Table 1). Over time, the concentration in the richest wealth quintile of adolescent non-mothers decreased (by -6·26%, 95%CI -6·85 to -5·67) and the concentration in the poorest of adolescent mothers increased (by 5·87%, 95% CI 4·74 to 7·00). The difference between the richest and poorest narrowed over time for the non-mothers (by 7.24%). For the mothers, the difference between richest and poorest widened by 12.22%, with an accelerated concentration of the mothers shifting down the wealth quintiles (22·66% to 28·53% of mothers in poorest households) and fewer mothers remaining in the richest quintiles (16·66% to 10·31% of mothers in the richest households over time) (Table 1 and S2 Fig).

Education levels increased for both mothers and non-mothers. Still, the increase was most significant for non-mothers. Over time, the fraction of non-mothers who attained secondary or higher education increased by 21·51% (95% CI 20·86–22·16) whereas for mothers, this increase was significantly lower at a 15·38% (14·29–16·47) increase. On an average, the

**Table 1. Distribution of social risk factors between adolescent mothers and non-mothers in LMICs.**

| Characteristics | Adolescent girls at baseline; % (n) | | | Adolescent girls at endline; % (n) | | | Total; % (n) | | | Difference (95% CI)* | |
|---|---|---|---|---|---|---|---|---|---|---|---|
| | Non-mother | Mother | Difference (95% CI) | Non-mother | Mother | Difference (95% CI) | Non-mother | Mother | Difference (95% CI) | Non-mother | Mother |
| Number of adolescent girls; N | 32,804 | 8,766 | | 62,987 | 15,410 | | 95,791 | 24,176 | | | |
| Wealth quintile | | | | | | | | | | | |
| • Poorest | 17·14 (5623) | 22·66 (1986) | 5·51(4·55, 6·48) | 18·12 (11411) | 28·53 (4396) | 10·41(9·64, 11·18) | 17·78 (17034) | 26·4 (6382) | 8·62(8·01, 9·22) | 0·98(0·47, 1·48) | 5·87(4·74, 7) |
| • Poorer | 16·54 (5426) | 21·07 (1847) | 4·53(3·59, 5·47) | 19·32 (12166) | 25·6 (3945) | 6·29(5·53, 7·04) | 18·36 (17592) | 23·96 (5792) | 5·59(5, 6·18) | 2·77(2·27, 3·28) | 4·53(3·43, 5·63) |
| • Middle | 17·84 (5851) | 20·93 (1835) | 3·1(2·15, 4·04) | 19·62 (12355) | 20·03 (3086) | 0·41(-0·29, 1·11) | 19·01 (18206) | 20·35 (4921) | 1·35(0·78, 1·91) | 1·78(1·26, 2·3) | -0·91 (-1·97, 0·15) |
| • Richer | 19·57 (6421) | 18·69 (1638) | -0·89(-1·81, 0·03) | 20·31 (12791) | 15·54 (2394) | -4·77(-5·42, -4·12) | 20·06 (19212) | 16·68 (4032) | -3·38(-3·91, -2·84) | 0·73(0·2, 1·27) | -3·15 (-4·15, -2·15) |
| • Richest | 28·91 (9483) | 16·66 (1460) | -12·25 (-13·17, -11·33) | 22·65 (14264) | 10·31 (1589) | -12·33 (-12·92, -11·75) | 24·79 (23747) | 12·61 (3049) | -12·18 (-12·68, -11·68) | -6·26 (-6·85, -5·67) | -6·34 (-7·26, -5·43) |
| Difference between poorest & richest; % | -11·77% | 6·00% | | -4·53% | 18·22% | | -7·01% | 13·79% | | | |
| Level of education | | | | | | | | | | | |
| • No education | 20·56 (6743) | 40·55 (3555) | 20(18·88, 21·12) | 10·13 (6380) | 24·94 (3844) | 14·82(14·09, 15·54) | 13·7 (13123) | 30·6 (7399) | 16·91(16·28, 17·53) | -10·43 (-10·92, -9·93) | -15·61 (-16·84, -14·38) |
| • Primary | 41·91 (13749) | 41·79 (3663) | -0·13(-1·29, 1·04) | 30·83 (19417) | 42·02 (6475) | 11·19(10·33, 12·05) | 34·62 (33166) | 41·93 (10138) | 7·31(6·62, 8) | -11·09 (-11·73, -10·44) | 0·23(-1·06, 1·53) |
| • Secondary or higher | 37·53 (12312) | 17·66 (1548) | -19·87 (-20·83, -18·92) | 59·04 (37190) | 33·04 (5091) | -26·01 (-26·84, -25·17) | 51·68 (49502) | 27·46 (6639) | -24·22 (-24·86, -23·57) | 21·51 (20·86, 22·16) | 15·38 (14·29, 16·47) |
| Difference between 'no education' and secondary or higher education | -16·97% | 22·89% | | -48·91% | -8·10% | | -37·98% | -3·14% | | | |
| Area of residence | | | | | | | | | | | |
| • Rural areas | 60·21 (19751) | 72·02 (6313) | 11·81(10·73, 12·89) | 59·11 (37229) | 72·24 (11132) | 13·13(12·33, 13·94) | 59·48 (56980) | 72·16 (17445) | 12·67(12·03, 13·32) | -1·1(-1·76, -0·45) | 0·22(-0·95, 1·4) |
| • Urban areas | 39·79 (13053) | 27·98 (2453) | -11·81 (-12·89, -10·73) | 40·89 (25758) | 27·76 (4278) | -13·13 (-13·94, -12·33) | 40·52 (38811) | 27·84 (6731) | -12·67 (-13·32, -12·03) | 1·1(0·45, 1·76) | -0·22(-1·4, 0·95) |
| Difference between rural an urban residence | 20·42% | 44·04% | | 18·22% | 44·48% | | 18·96% | 44·32% | | | |

*Absolute difference in the social risk factors of adolescent mother and non-mother between baseline and endline survey.

CI: Confidence interval.

proportion of adolescent mother without education decreased (by -15·61%, 95% CI: -16·84, -14·38) (Table 1)

There is a higher concentration of adolescent mothers in rural areas (72·24% at endline) than non-mothers (59·11% at endline), and this changed little over time (the change is not significant) (Table 1).

The national prevalence of adolescent motherhood decreased significantly for about half (9/20) of the studied countries between baseline (1996–2003) and endline (2014–2018) periods. The countries were Ethiopia, Guiana, Kenya, Malawi, Mali, Nigeria, Senegal, and Uganda

in sub-Saharan Africa (8/15 countries), and Haiti in other regions (1/5 countries). Except for Cambodia (where prevalent increased), rest of the studied countries (10/20), the prevalence of adolescent motherhood did not change significantly (Fig 1 and S1 Table).

In sub-Saharan Africa, of those countries where the prevalence decreased, the baseline prevalence was higher than for those countries where prevalence remained statistically unchanged. That is, in this region, baseline prevalence was greater than the global mean prevalence of the studied countries, 22% for 6/8 countries that experienced a decrease in adolescent motherhood and was greater than 22% for 2/8 countries that experienced no change in adolescent motherhood (Fig 1 and S1 Table).

The association between adolescent motherhood and the social risk factors has also changed over time. In the pooled analysis, adjusting for the three social risk factors and time trends (Model 3, Table 2), we see that over time poverty has increased in its association with adolescent motherhood. That is, over time, the poorest (AOR 1·42, 95% CI 1·28–1·59) and poorer (AOR 1·41, 95% CI 1·26–1·57) have significantly increased in the odds of their association with adolescent motherhood–even after controlling for education and rural living. As for education, no education (AOR 3·54, 95% CI 3·35–3·74) and primary education (AOR 2·05, 95% CI 1·96–2·14) are independently associated with higher odds of adolescent motherhood (Model 2, Table 2). Strikingly, the interaction with the time trend tells us that this association has not changed significantly over time (AOR = 0·92, 95%CI 0·84, 1·01 for no education) (Model 3, Table 2). As for rural residents, while in the unadjusted model (Model 1), rural living is strongly associated with adolescent motherhood (OR 1·89, 95% CI 1·82–1·96). Once we control for wealth and education (AOR 1·05, 95% CI 1·01–1·10; Model 2), and for time trends (AOR 1·09, 95% CI 1·01–1·17; Model 3), the association weakened but remained statistically significant in these adjusted models (Table 2).

At the country level, we observed that changes in the social risk factors of adolescent motherhood varied across countries. The poorest are more likely to be increasingly associated with adolescent motherhood over time in Ethiopia, Chad, Nigeria, Cambodia, Zimbabwe, and Malawi, in the face of decreases in the overall prevalence in adolescent motherhood in Ethiopia, Nigeria and Malawi in this group of countries. On the contrary, in Mali, Benin, and the Philippines, the poorest are less likely to be associated with adolescent motherhood (Fig 2). For 11 of the 20 countries (55%), the association between wealth and adolescent motherhood has not changed over time. At the country level, the trend in the association of rural residence has a similar trend to wealth (Fig 2).

At the country level, in terms of the association of education with adolescent motherhood, no education is increasingly associated with adolescent motherhood in Ethiopia and Cambodia. However, in Uganda, Guinea and Benin, the strength of association between no education and adolescent motherhood is lessening over time. Association with no education remaining unchanged during the study period in rest 75% (15/20) of the countries (Fig 2).

## Discussion

This study aimed to examine transition in social risk factors for adolescent mothers over time and identify diverging and converging differences in social risk factors for them in LMICs. The study also aimed to examine the association of the social factors and adolescent motherhood in LMICs and how this association changed over time. Our study explored that social risk factors of the adolescent mother had shifted in different directions during MDGs and SDGs eras, and adolescent mother remained more disadvantaged than non-mother in LMICs. The strength of association between adolescent motherhood and low socioeconomic status, and rural residence increased significantly; however, the strength of association between

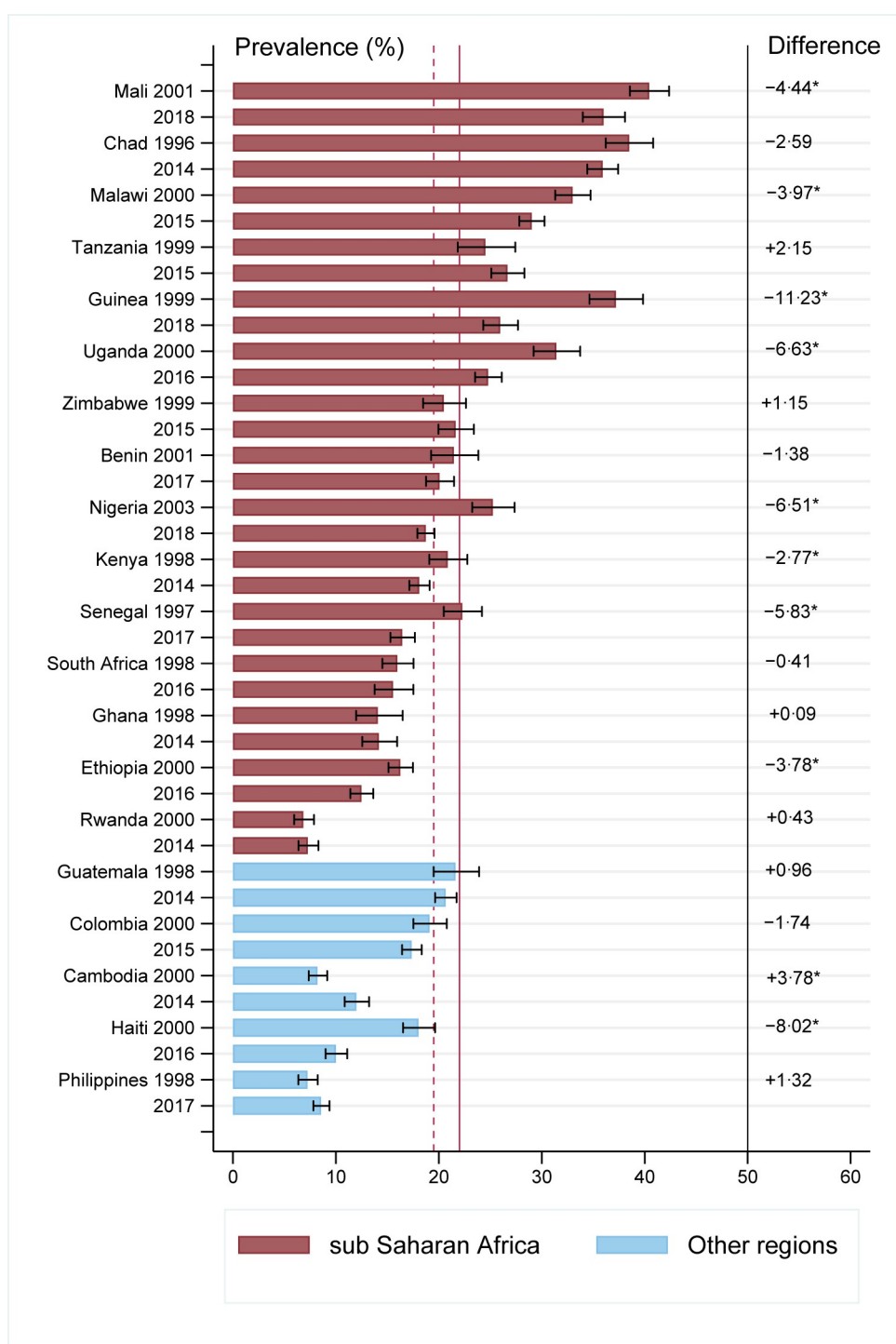

**Fig 1. Transition in the national level prevalence of adolescent motherhood in LMICs, 1996–2018 (countries are ranked by endline survey prevalence).** *Statistically significant. The solid and short-dash lines represent the global mean prevalence of adolescent motherhood in the study countries, at baseline and endline, respectively. Negative and positive differences represent reduction and increment in the prevalence of adolescent motherhood, respectively.

**Table 2. Association between social factors and adolescent motherhood in LMICs (AOR): Pooled estimates.**

| Social factors | Model 1: Unadjusted | Model 2: Adjusted | Model 3: Adjusted (with interaction) |
|---|---|---|---|
| | OR (95% CI) | AOR (95% CI) | AOR (95% CI) |
| Wealth quintile | | | |
| • Poorest (vs Richest) | 3·53(3·35, 3·73) | 2·20(2·07, 2·35) | 1·79(1·63, 1·96) |
| • Poorer (vs Richest) | 2·98(2·82, 3·15) | 2·07(1·95, 2·21) | 1·69(1·54, 1·86) |
| • Middle (vs Richest) | 2·36(2·23, 2·49) | 1·80(1·69, 1·91) | 1·67(1·53, 1·83) |
| • Richer (vs Richest) | 1·73(1·63, 1·83) | 1·46(1·38, 1·55) | 1·46(1·34, 1·60) |
| Wealth quintile X Survey time[1] | | | |
| • Poorest X Endline survey | – | – | 1·42(1·28, 1·59) |
| • Poorer X Endline survey | – | – | 1·41(1·26, 1·57) |
| • Middle X Endline survey | – | – | 1·15(1·03, 1·29) |
| • Richer X Endline survey | – | – | 1·01(0·91, 1·13) |
| Level of education | | | |
| • No education (vs Secondary or higher) | 4·67(4·43, 4·92) | 3·54(3·35, 3·74) | 3·72(3·42, 4·04) |
| • Primary (vs Secondary or higher) | 2·46(2·36, 2·56) | 2·05(1·96, 2·14) | 2·00(1·86, 2·15) |
| Level of education X Survey time[1] | | | |
| • No education X Endline survey | – | – | 0·92(0·84, 1·01) |
| • Primary X Endline survey | – | – | 1·05(0·96, 1·14) |
| Area of living | | | |
| • Rural areas (vs Urban area) | 1·89(1·82, 1·96) | 1·05(1·01, 1·10) | 1·04(0·98, 1·12) |
| Area of residence X Survey time[1] | | | |
| • Rural areas X Endline survey | – | – | 1·09(1·01, 1·17) |

[1]X represents an interaction. AOR from the interaction term in Model 3 tells transition in the social risk factor of adolescent motherhood from baseline to endline. AOR>1 (<1) for the interaction terms in Model 3 represents increased (decreased) in the association between adolescent mother and social risk factors at endline compared to baseline.

OR: Odds ratio; AOR: Adjusted odds ratio; CI: Confidence interval.

adolescent motherhood and low level of education remained unchanged in LMICs since the MDGs started.

Efforts to improve adolescent girls' sociodemographic profile during the MDGs and SDGs eras were made, particularly in terms of poverty reduction and improved education. However, we observed that the adolescent mothers remain disadvantaged. Over time, the overall risk of adolescent motherhood among the poorest adolescent girls and the adolescent girls who live in rural areas had increased. Several studies support our study findings. Neal et al. did two comprehensive studies in east Africa and Latin America and the Caribbean regions. They reported that adolescent first birth continued to be more common among the poorest and rural residents [23, 35]. Another evaluation study in Malawi from 2004 to 2016, showed that the socio-economic inequality in teenage pregnancy and childbearing worsened to the disadvantage of the adolescent girls from poorer backgrounds [36]. Therefore, our observation and other evidence indicate that adolescent mothers are likely to become increasingly concentrated amongst the poor over time, which may create further marginalisation and disadvantage for the adolescent girls in LMICs.

Another social risk factor associated with adolescent motherhood in our study was adolescent girl's level of education. In our study, adolescent girls' secondary or higher education was associated with a lower level of adolescent motherhood. It is well documented in both developed and developing countries that girls' higher education level is a protective factor for early pregnancy [25]. With the analysis of adolescent motherhood and education over time, we

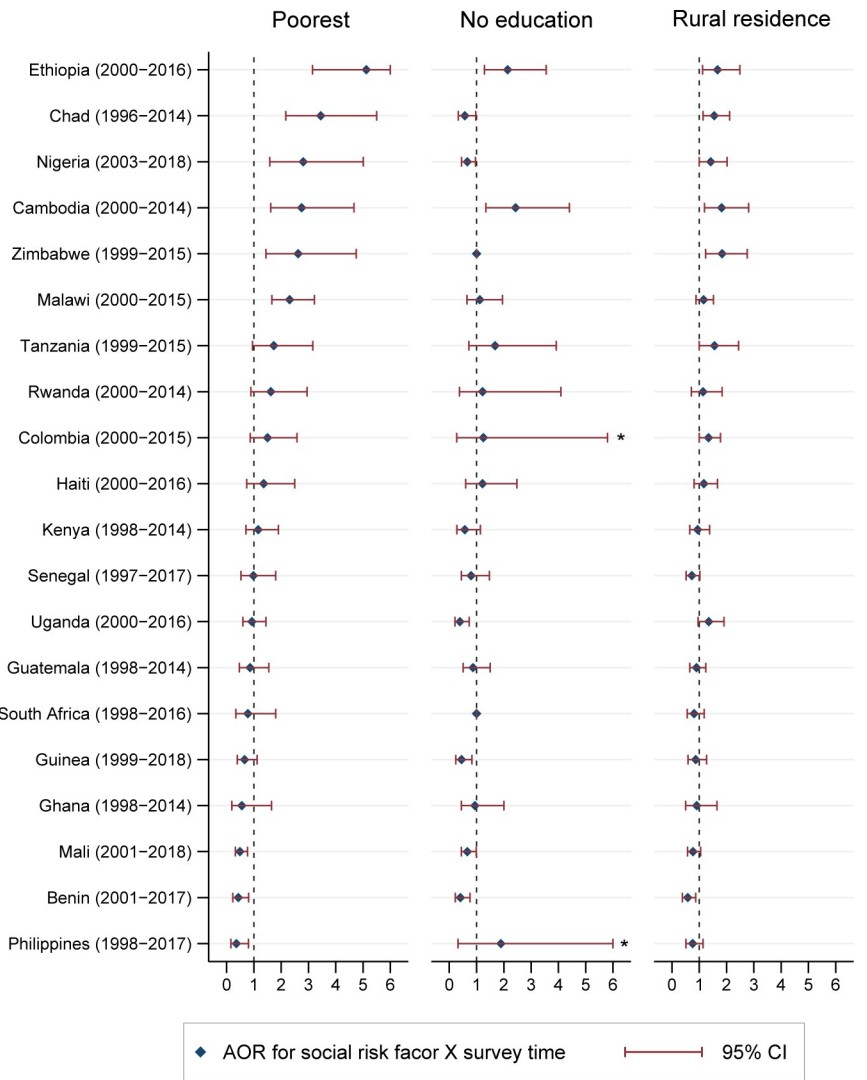

**Fig 2. Transition in the social risk factor of adolescent motherhood in LMICs (AOR) during 1996–2018: Country specific estimates.** *Value of upper limit of the 95% CI is > 6.0. AOR: Adjusted Odds Ratio; CI: Confidence Interval. Only coefficient of interaction terms for 'wealth quintile' (ref: richest), 'level of education' (ref: secondary or higher) and 'area of resident' (ref: urban residence) are presented in this figure. X represents interactions. AOR>1 (<1) represents increased (decreased) in the association between adolescent mother and social risk factors at endline compared to baseline.

showed that this association between adolescent motherhood and their education level has not changed over time. Girls with no education have seen no change in their risk of adolescent motherhood over time. While we have known the risk of no education has in fuelling adolescent motherhood; nothing has been effective in dampening this association. Through the education association lens, programs could be effective in reducing adolescent motherhood by increasing level of education. With fewer girls with lower education, there would be the follow-on effect of fewer girls at risk of adolescent motherhood.

Our results showed that level of education increased for adolescent girls in the studied countries. This increase in education may have had the trickle-down effect of reducing adolescent motherhood prevalence observed in several studied countries. Yet on average, the association of low education and adolescent motherhood did not change significantly over time

(Table 2) and only weakly reduced for Uganda, Guinea and Benin in the 20 studied countries (Fig 1). While level of education improved for many adolescent girls, their risk of adolescent motherhood remained unchanged for those who were left behind with low education. For this disadvantaged group, an effective program needs to be identified to reduce the risk of pregnancy. For example, the active labour market program (ALMP) in the Dominican Republic that aimed to improve the labour market entry of disadvantaged youth effectively reduced teenage pregnancy among disadvantaged youth [37].

During MDGs and SDGs eras, the prevalence of adolescent motherhood has declined in less than half of the studies. The prevalence remains over 20% in 9/20 studies countries, indicating the progress in reducing adolescent motherhood was slow or stagnant in studied countries in the last 15 years. The slow declined in adolescent motherhood are consistent with those from other studies in LMICs [2, 23]. Huda et al. (2020) assessed the time trend in adolescent motherhood, including 74 LMICs, and concluded that the rate of reduction in the prevalence of adolescent motherhood was either slow or absent in many LMICs [2]. Another study conducted in five countries in the Latin America and the Caribbean regions, which describe the little progress in reducing adolescent first birth over the last two decades [23]. This lack of progress is a concern for achieving the global goal, which needs to be taken care of.

The social risk factors of poor health produce widespread inequities in the prevalence of adolescent motherhood within and between countries and can be a severe obstacle in reducing the burden of adolescent motherhood in LMICs. Our findings on the increased trend of disadvantaged (poorest) adolescent motherhood and the persistent association of low education with adolescent motherhood indicate a lack of effective policy and program for that disadvantaged group. Designing more effective intervention by targeting this most disadvantaged group could be beneficial for controlling this upward trend of inequalities [38]. Programs and policies that improve the sociodemographic profile create more opportunities for education, provide proper access to sexual and reproductive health care can mitigate the inequalities and eventually that will prevent adolescent motherhood in LMICs.

Our study is not without its limitations. First, our analysis of repeated cross-sectional data does not fully understand the mechanism, causal pathway, and mediating factor. Secondly, as per WHO definition, the adolescent age group should be 10–19 years old; however, the DHS collects only data from 15–45 years of women; therefore, we were unable to include the early adolescents (10–14 years old) in our analysis. Thirdly, unlike the measurement of the level of education and areas of residence, the wealth quintile is a household measure of wealth relative to the wealth of other households within the survey and is only a proxy measure of adolescent girls' socioeconomic status. Thus, the wealth quintile measured at baseline and endline survey may not give us the appropriate trend of adolescent girls with a particular level of wealth quintile. To overcome this limitation, in addition to examine the changes in a specific quintile (poorest), we also examined the changes in the difference between first (poorest) and last (richest) quantile during the study period in this study. Fourth, accuracy of the age of the participants at the time of interview and the year of their first birth can be affected by recall bias, which might have introduced bias into the prevalence estimate. Fifth, our estimates included livebirths and current pregnancies, but did not include stillbirths, miscarriages, or abortions. Data on stillbirths, miscarriages, or abortions are not consistently reported in all the countries. Thus, in this study, the focus is on motherhood rather than pregnancy. Finally, some of the exposure variables were measured at the time of the interview; however, the outcome variable (motherhood) was measured based on birth history data and may not align in the exposure-outcome pathway.

In conclusion, we found that only half of the studied countries have experienced a decrease in the prevalence of adolescent motherhood over time. Social determinants of adolescent

mother have shifted in different directions over time for mothers and non-mothers. The association of low socioeconomic status with adolescent motherhood increased significantly; however, its association with a low level of education remained unchanged in LMICs since the MDGs started. Furthermore, for social risk factors that have long been known (education) with no change in the association with adolescent motherhood over time, no programmatic efforts have been effective in improving the social risk factors of adolescent mothers' faces. These results indicate that efforts need to be enhanced to improve adolescent girl's education in LMICs. More attention should bring to the disadvantaged adolescent girls, such as the poorest who lives in rural areas, continued improvements in education, and programs that reduce the risk of pregnancy for girls with low education, for prioritising interventions to accelerate the reduction in the prevalence of adolescent motherhood in LMICs.

## Supporting information

**S1 Fig. Analytic sample in this study.** *The sample selection criterion was, country with at least two all-women surveys: a survey in 1996–2003, near to MDGs started (as baseline) and another survey in 2014–2018, recent time (as endline) so that the duration for assessing the transition of social factors can be nearly 15 years for all the studied countries.
(TIF)

**S2 Fig. Absolute difference between the proportion of first and last categories of the social risk factors among adolescent mother and non-mothers at baseline and endline.**
(TIF)

**S1 Table. List of survey countries, study participants (adolescent girls aged 15–19 years), weighted prevalence of adolescent motherhood.** *Difference in the prevalence between baseline and endline survey; CI: Confidence interval.
(DOCX)

**S2 Table. Measurement of variables used in this study.**
(DOCX)

**S3 Table. Association between social factors and adolescent motherhood in LMICs, and changes in the association (AOR) during 1996–2018: Country-specific estimates.** AOR: Adjusted odds ratio; CI: Confidence interval; AOR>1 (<1) represents increased (decreased) in the association between adolescent mother and social risk factors at endline compared to baseline.
(DOCX)

## Acknowledgments

An Australian Government Research Training Program (RTP) Scholarship supported this research. We would like to acknowledge with gratitude the commitment of the Australian Government and The University of Queensland to its research efforts.

## Author Contributions

**Conceptualization:** M. Mamun Huda, Abdullah Al Mamun.

**Data curation:** M. Mamun Huda.

**Formal analysis:** M. Mamun Huda, Jocelyn E. Finlay, Martin O'Flaherty.

**Methodology:** M. Mamun Huda, Jocelyn E. Finlay, Martin O'Flaherty.

**Supervision:** Jocelyn E. Finlay, Martin O'Flaherty, Abdullah Al Mamun.

**Writing – original draft:** M. Mamun Huda.

**Writing – review & editing:** M. Mamun Huda, Jocelyn E. Finlay, Martin O'Flaherty, Abdullah Al Mamun.

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
