## [Decision Letter · Decision Letter 0]

29 Oct 2021

PGPH-D-21-00062

Transition in social risk factors and adolescent motherhood in low- and middle-income countries: Evidence from Demographic and Health Survey data, 1996-2018

Dear Dr. Huda,

Thank you for submitting your manuscript to PLOS Global Public Health. After careful consideration, we feel that it has merit but does not fully meet PLOS Global Public Health’s publication criteria as it currently stands. Therefore, we invite you to submit a revised version of the manuscript that addresses the points raised during the review process.

Reviewer 2 has provided suggestions on how this manuscript could be improved. Please consider these suggestions and we invite you to submit a revised version of the manuscript.  

We look forward to receiving your revised manuscript.

Kind regards,

Rebecca M Flueckiger

Academic Editor

Journal Requirements:

1. Please list the name and version of any software package used for statistical analysis, alongside any relevant references. For more information on PLOS ONE's expectations for statistical reporting, please see https://journals.plos.org/plosone/s/submission-guidelines.#loc-statistical-reporting.

2. Please update the completed 'Competing Interests' statement, including any COIs declared by your co-authors. If you have no competing interests to declare, please state "The authors have declared that no competing interests exist". Otherwise please declare all competing interests beginning with the statement "I have read the journal's policy and the authors of this manuscript have the following competing interests:"

Additional Editor Comments (if provided):

Reviewers' comments:

Reviewer's Responses to Questions

**Comments to the Author**

1. Does this manuscript meet PLOS Global Public Health’s publication criteria? Is the manuscript technically sound, and do the data support the conclusions? The manuscript must describe methodologically and ethically rigorous research with conclusions that are appropriately drawn based on the data presented.

Reviewer #1: Yes

Reviewer #2: Partly

2. Has the statistical analysis been performed appropriately and rigorously?

Reviewer #1: Yes

Reviewer #2: Yes

3. Have the authors made all data underlying the findings in their manuscript fully available (please refer to the Data Availability Statement at the start of the manuscript PDF file)?

Reviewer #1: Yes

Reviewer #2: Yes

4. Is the manuscript presented in an intelligible fashion and written in standard English?

Reviewer #1: Yes

Reviewer #2: No

5. Review Comments to the Author

Reviewer #1: The manuscript titled "Transition in social risk factors and adolescent motherhood in low-and middle-income countries: Evidence from Demographic and Health Survey data , 1996-2018" is excellent work to describe changes of socioeconomic factors of the adolescent motherhood in developing countries. Hope findings of the study will have significant impact on prioritising women and adolescent health and taking action on reducing inequities in SDG eras.

I have no comments.

Please provide explains full meaning of abbreviation in line 86., SRHR and line 133- ICF

Reviewer #2: The study explores association between adolescent motherhood and the social risk factors, as well as changes in the strength of the association over time in 20 LMICs using DHS data. The results constitute a significant addition to the field. However, a few methodological issues must be addressed before the manuscript is suitable for publication. Also, the authors could potentially benefit from editorial services. My comments on specific lines of the paper are as follows.

Lines 85-88: lack of sufficient interventions from the governments may be one of the reasons why high levels of adolescent motherhood have been observed in many LMICs but should not be the only reason. The authors should consider doing a more comprehensive review or refining the wording.

Lines 89-106 discuss the risk factors of adolescent pregnancy at the individual level. The paper could be strengthened by including an overview of the risk factors at other levels, such as community and society levels.

Line 89: the authors introduce the notion “social risk factors” here. Would be good if the concept can be defined and a brief discussion on social risk factors vs. non-social risk factors can be provided.

Lines 130-131: The authors indicate that data from 20 LMICs are used in the analysis. However, it is unclear what the selection criteria are.

Lines 136-137: Would be good if the authors can clarify what regions are they - are these SDG regions?

Lines 152-158; 172-173: The authors assesse changes in household socioeconomic status using the household wealth index. However, the DHS wealth index is intended to compare households within a single survey. Scores are relative, not absolute; in any particular survey, the mean wealth score is 0. As a country's population's economic status improves and household services (such as piped drinking water) and household assets (such as televisions) become more affordable, the combination of assets and services that ranks a household in the top quintile in an earlier survey may rank a household in the bottom quintile in a later survey. See further details in references below. The implications and limitations of using the wealth index must be discussed in addition to lines 361-366.

References:

Staveteig, Sarah and Lindsay Mallick. 2014. Intertemporal Comparisons of Wealth with DHS Data: A Harmonized Asset Index Approach. DHS Methodological Reports No. 15. Rockville, Maryland, USA: ICF International.

Rutstein, Shea O. and Kiersten Johnson. 2004. The DHS Wealth Index. DHS Comparative Reports No. 6. Calverton, Maryland: ORC Macro.

Lines 185: Suggest to the authors to further elaborate “cluster/cluster levels”.

Lines 201-202: The statement does not seem to be accurate considering improved education and urbanization.

Lines 205-212: Further elaborations on “gaps between the richest and the poorest” can be helpful. The authors may wish to explore the measures “% mothers/non-mothers among poorest/richest”.

Lines 224-229: When discussing changes between the two surveys, it would be helpful to know the intervals between the surveys. If the intervals have a large range, the authors should discuss the implications and consider taking this factor in the analysis. In addition, the authors should consider if additional surveys were done between the baseline and endline surveys, and offer an explanation for why these surveys were not included in the study.

Lines 238-240: The rationale for a threshold of 22% is unclear. Would be great if the authors could clarify.

Line 335: Typo – “studies” should be “countries”

Line 347 “Our findings on the increased trend of disadvantaged (poorest and living in a rural area) adolescent motherhood…” seems to be inconsistent with lines 217-219 “There is a higher concentration of adolescent mothers in rural areas (72·24% at endline) than non-mothers (59·11% at endline), and this changed little over time (the change is not significant)”. Would be good if the authors can clarify.

Lines 352-356: The paper could benefit from a more elaborated discussion on these points instead of a general statement.

Lines 368-370: Common quality issues on pregnancy/birth data in DHS should also be discussed.

Lines 376-377: The statement “These results indicate that little has been done to improve the prevalence of adolescent motherhood in LMICs…” does not seem to be accurate. The results show that half of the countries in the study have experienced a decrease in the prevalence of adolescent motherhood. Also, much work has been done in the field, the lack of progress in some countries might be attributed to a variety of factors. Absolute assertions like these should be avoided by the authors.

6. PLOS authors have the option to publish the peer review history of their article (what does this mean?). If published, this will include your full peer review and any attached files.

**Do you want your identity to be public for this peer review?** For information about this choice, including consent withdrawal, please see our Privacy Policy.

Reviewer #1: **Yes: **Ganchimeg Togoobaatar

Reviewer #2: **Yes: **Mengjia Liang

---

## [Decision Letter · Decision Letter 1]

13 Apr 2022

Transition in social risk factors and adolescent motherhood in low- and middle-income countries: Evidence from Demographic and Health Survey data, 1996-2018

PGPH-D-21-00062R1

Dear Mr. Huda,

We are pleased to inform you that your manuscript 'Transition in social risk factors and adolescent motherhood in low- and middle-income countries: Evidence from Demographic and Health Survey data, 1996-2018' has been provisionally accepted for publication in PLOS Global Public Health.

Best regards,

Rebecca M Flueckiger, Ph.D.

Academic Editor

Reviewer Comments (if any, and for reference):

Reviewer's Responses to Questions

**Comments to the Author**

1. If the authors have adequately addressed your comments raised in a previous round of review and you feel that this manuscript is now acceptable for publication, you may indicate that here to bypass the “Comments to the Author” section, enter your conflict of interest statement in the “Confidential to Editor” section, and submit your "Accept" recommendation.

Reviewer #2: All comments have been addressed

2. Does this manuscript meet PLOS Global Public Health’s publication criteria? Is the manuscript technically sound, and do the data support the conclusions? The manuscript must describe methodologically and ethically rigorous research with conclusions that are appropriately drawn based on the data presented.

Reviewer #2: Yes

3. Has the statistical analysis been performed appropriately and rigorously?

Reviewer #2: Yes

4. Have the authors made all data underlying the findings in their manuscript fully available (please refer to the Data Availability Statement at the start of the manuscript PDF file)?

Reviewer #2: Yes

5. Is the manuscript presented in an intelligible fashion and written in standard English?

Reviewer #2: Yes

6. Review Comments to the Author

Reviewer #2: No further comments

7. PLOS authors have the option to publish the peer review history of their article (what does this mean?). If published, this will include your full peer review and any attached files.

**Do you want your identity to be public for this peer review?** For information about this choice, including consent withdrawal, please see our Privacy Policy.

Reviewer #2: **Yes: **Mengjia Liang
